# Micronutrients Deficiencies in 374 Severely Malnourished Anorexia Nervosa Inpatients

**DOI:** 10.3390/nu11040792

**Published:** 2019-04-05

**Authors:** Mouna Hanachi, Marika Dicembre, Claire Rives-Lange, Jacques Ropers, Pauline Bemer, Jean-Fabien Zazzo, Joël Poupon, Agnès Dauvergne, Jean-Claude Melchior

**Affiliations:** 1Nutrition Unit, Raymond Poincaré University Hospital (Assistance Publique–Hôpitaux de Paris), 92380 Garches, France; marika.dic@libero.it (M.D.); pauline.bemer@aphp.fr (P.B.); jfzazzo@hotmail.fr (J.-F.Z.); jean-claude.melchior@aphp.fr (J.-C.M.); 2Versailles Saint Quentin-en-Yvelines University, 78180 Montigny-le-Bretonneux, France; 3Nutrition Unit, Georges Pompidou University Hospital (Assistance Publique–Hôpitaux de Paris), 75015 Paris, France; claireriveslange@hotmail.fr; 4Paris Descartes University, 75006 Paris, France; 5Clinical Research Unit, Pitié-Salpêtrière University Hospital, (Assistance Publique–Hôpitaux de Paris, 75013 Paris, France; jacques.ropers@aphp.fr; 6Laboratory of Biological Toxicology, Saint Louis–Lariboisiere, University Hospital Paris France, 75010 Paris, France; joel.poupon@aphp.fr; 7Laboratory of Biochemistry, Beaujon University Hospital (Assistance Publique–Hôpitaux de Paris), 92110 Clichy, France; agnes.dauvergne@aphp.fr

**Keywords:** anorexia nervosa, malnutrition, micronutrients, zinc, copper, selenium, vitamin B1, vitamin B12, vitamin D, vitamin B9

## Abstract

Introduction: Anorexia nervosa (AN) is a complex psychiatric disorder, which can lead to specific somatic complications. Undernutrition is a major diagnostic criteria of AN which can be associated with several micronutrients deficiencies. Objectives: This study aimed to determinate the prevalence of micronutrients deficiencies and to compare the differences between the two subtypes of AN (restricting type (AN-R) and binge-eating/purging type (AN-BP)). Methods: We report a large retrospective, monocentric study of patients hospitalized in a highly specialized nutrition unit between January 2011 and August 2017 for severe malnutrition treatment in the context of anorexia nervosa. Results: Three hundred and seventy-four patients (360 (96%) women, 14 (4%) men), age: 31.3 ± 12.9 years, Body Mass Index (BMI): 12.5 ± 1.7 kg/m^2^ were included; 253 (68%) patients had AN-R subtype while, 121 (32%) had AN-BP. Zinc had the highest deficiency prevalence 64.3%, followed by vitamin D (54.2%), copper (37.1%), selenium (20.5%), vitamin B1 (15%), vitamin B12 (4.7%), and vitamin B9 (8.9%). Patients with AN-BP type had longer disease duration, were older, and had a lower left ventricular ejection fraction (LVEF) (*p* < 0.001, *p* = 0.029, *p* = 0.009), when compared with AN-R type, patients who instead, had significantly higher Alanine Aminotransferase (ALT) and Brain Natriuretic Peptide (BNP) levels (*p* < 0.001, *p* < 0.021). In the AN-BP subgroup, as compared to AN-R, lower selenium (*p* < 0.001) and vitamin B12 plasma concentration (*p* < 0.036) were observed, whereas lower copper plasma concentration was observed in patients with AN-R type (*p* < 0.022). No significant differences were observed for zinc, vitamin B9, vitamin D, and vitamin B1 concentrations between the two types of AN patients. Conclusion: Severely malnourished AN patients have many micronutrient deficiencies. Differences between AN subtypes are identified. Micronutrients status of AN patients should be monitored and supplemented to prevent deficiencies related complications and to improve nutritional status. Prospective studies are needed to explore the symptoms and consequences of each deficiency, which can aggravate the prognosis during recovery.

## 1. Introduction

Anorexia Nervosa (AN) is an eating behavior disorder that affects 1–2.2% [1,2] of young women. The important disease burden is reflected by a higher mortality rate, up to 12 times, as compared to general population [3]. According to the fifth edition of Diagnostic and Statistical Manual of Mental Disorders (DSM-V), AN is defined as an inability to maintain adequate dietary intakes, to maintain normal weight for age, associated with an intense fear of gaining weight or becoming fat, or persistent behavior that interferes with weight gain and disturbances of the body image [4]. Two sub-types of AN are described, the pure restricting sub-type (AN-R), and the binge-eating/purging sub-type (AN-BP), with cycles of large meals that are followed by purging behaviors (vomiting and/or laxative abuse) [4]. Anxiety and depressive symptoms and problematic physical activity (PPA) are associated to this symptomatology [5,6]. Somatic complications, along with suicide, are the two first mortality causes [7]. AN somatic complications can be the consequence of either undernourishment, pathological behaviors that are associated with food restriction (vomiting, potomania, PPA) or refeeding adverse effects. The somatic complications of undernutrition in AN patients are well described in the literature [8]. Cardiac failure risk [9], hypertransaminasemia and hepatic failure [10], functional intestinal disorders, such us bloating and constipation [11], hematological disturbances [12], bone demineralization, and hormonal disorders are the most frequently reported complications [13]. Malnutrition in AN is most often of marasmus type (a proteino-energetic chronic adaptive undernutrition), more rarely comprising a kwashiorkor [14,15]. The prevalence of micronutrients deficiencies and their consequences in malnourished patients with AN are poorly known. Indeed, few studies focused on patients with moderate undernutrition (Mean BMI > 15) and evaluated micronutrients deficiencies in AN patients, with conflicting results. In fact, some clinical and biological symptoms that were observed in AN could be partially explained by micronutrients deficiencies. For example, Suzuki and colleagues described an association between zinc deficiency and restrictive eating behavior [16]; cases of sensory neuropathy were found in AN patients with vitamin B12 deficiency [17]; fasting can cause neurological complications due to severe vitamin B1 deficiency [18]. Moreover differences between AN sub-types may be expected, purging behaviors in AN-BP, such as vomiting and laxative misuse, could lead to more severe and different micronutrients deficiencies.

Although micronutrient body reserves decrease during chronic undernutrition phase in patients with AN, the deficiencies remain mostly asymptomatic at this stage; micronutrient needs remain limited because of the physiological adaptation to prolonged fasting or semi-fasting situations. Clinical manifestations reveal these deficiencies, especially during renutrition times due to the increase of metabolic needs [19]. An assessment of micronutrient status before re-feeding malnourished AN patients should be a prerequisite in order to anticipate the occurrence of symptomatic deficiencies that may complicate patient’s care management along with his prognosis through individualized supplementation. International recommendations remain imprecise on the modalities of micronutrient supplementation in AN patients and need to be updated in the light of new studies [20,21]. The present study aims to determine the prevalence of asymptomatic micronutrient deficiencies (vitamins and trace elements) in a large cohort of severely malnourished AN patients that were hospitalized in a national referral center of clinical nutrition and eating disorders. 

For this study, all micronutrients that were dosed routinely for AN inpatients at admission were included; in our unit: vitamin B1 (thiamin), B12 (cobalamin), B9 (folates), D (vitamin D3, cholecalciferol), zinc, copper, and selenium. In addition to micronutrient dosage, other parameters have been explored to facilitate the interpretation of micronutrient deficits: for example, (1) C reactive protein (CRP) to detect inflammation, which can influence selenium, vitamins B9, and B1 levels; (2) Brain Natriuretic Peptide (BNP) and left ventricular ejection fraction (LVEF), markers of heart failure, may be associated with selenium deficiency; and (3) GGT and alkaline phosphatase are markers of liver cholestasis that can alter copper blood level.

## 2. Material and Methods

### 2.1. Patients and Study Design

A large retrospective, unicenter study was carried out. All of the consecutive undernourished inpatients with AN (DSM - IV or V) [4,22] that were referred to the Clinical Nutrition Department in Raymond Poincaré University Hospital (Garches, France) between January 2011 and August 2017 were included. For patients that were hospitalized more than one time, the date of first hospitalization was the date of inclusion in the study. We excluded all the patients with concomitant comorbidity that could affect micronutrient blood levels as such coeliac disease, digestive tract disease, cancer, alcoholism, and diabetes. All of the data were recovered from the hospital medical records.

### 2.2. Ethical Aspects

The study was conducted in accordance with the relevant French guidelines and regulations. Regular mailing sent individual information to all patients. Patient non-opposition was a prerequisite for the use of their data. The study was declared to the French data protection authority (CNIL: 2029030vO).

### 2.3. Measures and Procedure

The following clinical and biological data were collected at admission and before starting nutritional care: disease duration (years), type of anorexia nervosa according to DSM V [4] or and DSM-IVr [22], height (meter), and weight at admission (kilogram), which were measured in the morning, in light clothes without shoes. The Body Mass Index (BMI) was calculated as body weight divided by squared height (kg/m^2^). After a 12h fasting period, the biochemical analyses were performed from venous blood sample in the hospital’s central laboratory with routine methods, while using reference measurements from the Department of Clinical Biochemistry. Using a large sample among adult healthy population determined the normal values of the laboratory. Micronutrients were assayed in two distinct labs according micronutrients. As such, zinc, copper, and selenium were assayed at the Biological Toxicology Laboratory, Saint Louis–Lariboisiere University Hospital, Paris (France), while vitamin B1 was assayed at the Biochemistry Laboratory of Beaujon University Hospital, Clichy La Garenne (France).

Left ventricular ejection fraction (LVEF) was assessed to evaluate cardiac left ventricular function by echocardiography for every each patient. Finally, Dual-energy X-ray absorptiometry (DEXA) was performed and fat mass was collected for this study.

### 2.4. Statistical Analysis

The results were analyzed using R software (version 3.5.1. 2018-07-02, x86_64-pc-linux-gnu, R Foundation for Statistical Computing, Vienna, Austria). The Wilcoxon rank-sum test and Fisher test were used for comparing two groups on continuous and categorical variables, respectively. A *p* value that was lower than 0.05 was considered to be statistically significant.

## 3. Results

### 3.1. Patients Characteristics

Three hundred ninety-four patients were hospitalized on the selected period, 17 refused to participate. Ultimately, 374 were included in the study, while three patients overall were excluded for Coeliac (*n* = 2) and Crohn disease (*n* = 1). Study population includes 360 (96.3%) female and 14 (3.7%) male patients, with a mean age of 31.3 ± 12.9 and 253 (68%) patients out of 374 had a restrictive AN (AN-R) subtype, while 121 (32%) a binge-purging (AN-BP) subtype. All of the patients were severely malnourished, with mean weight and BMI, of 33.7 ± 5.9 kg and 12.5 ± 1.7 kg/m^2^ (Table 1). 

Patients with AN-BP sub-type had longer disease duration, were older, and had a lower LVEF (*p* < 0.001, *p* < 0.029, *p* < 0.009, respectively). Patients with AN-R subtype had a higher ALT and BNP (*p* < 0.001, *p* < 0.021).

### 3.2. Prevalence of Micronutrients Deficiencies

The prevalence of micronutrient deficiencies in the combined population was: For trace elements: 64.3% for zinc (<12.5 µmol/L), 37.1% for copper (<12.7 µmol/L), and 20.5% for selenium (<0.9 µmol/L). For Vitamins: 54.2% for vitamin D (<30 ng/mL), 15% for vitamin B1, 4.7% (<126 ng/L) for vitamin B12 (<197 ng/L), and 8.9% for vitamin B9 (<3.90 µg/L). Figure 1 presents the distribution of each micronutrient. The majority of patients (92.8%) had at least one detected micronutrient deficiency (Table 2).

Comparative results between the two AN subgroups showed some differences in micronutrients status. Indeed, in the AN-BP subgroup, we observed lower selenium (*p* < 0.001) and vitamin B12 plasma concentrations (*p* < 0.036), while in the AN-R, a lower concentration of copper (*p* < 0.022) was detected in plasma blood. No significant differences were observed for zinc, vitamin B9, vitamin D, and vitamin B1 in both AN subgroups (Table 3). 

Additionally, no association was found between LVEF, BNP, and micronutrients deficiency, and though mean LVEF was in the normal range for the two AN subtypes, a significantly lower LVEF in the AN-BP sub-type was found (*p* = 0.009) (Table 1).

## 4. Discussion

To our knowledge, this study is the largest published on micronutrient status in severely malnourished adult AN patients with an average BMI that was lower than 13. The results showed that the majority of patients had at least one micronutrient deficiency (Table 3). Zinc was the most frequent trace element deficiency (64.3%), whereas 37% and 21% of patients exhibited copper and/or selenium deficiencies, respectively. The two most frequent vitamin deficiencies concerned vitamin D and vitamin B1 for 54.2% and 15% of patients, respectively. When comparing the two subtypes of eating disorders, the selenium and vitamin B12 levels were lower in AN-BP patients, while copper level was lower in AN-R. These results are the consequence of restrictive macro and micronutrients food intakes, largely described in AN patients [23,24]. 

Despite a low BMI, albumin and transthyretin, which are the usual nutritional protein markers, were in a normal range for 88% of AN patients. Indeed, AN patients experience an adaptive chronic malnutrition, leading to an extended semi-fasting marasmus undernutrition type [14]. 

Thirty-five per cent of our patients presented hypertransaminasemia (AST and/or ALT ≥ 2N) at admission; these results are in accordance with our past findings (10). Liver enzymes (ALT) were higher in AN-R; it is probably related, after the exclusion of other potential liver or non-liver diseases, to the fact that total fasting in AN restricted type patients led to deep glycogenic depletion and the autophagy of hepatocytes [25].

Patients with AN-R type were younger and they had shorter disease duration; this can be explained by the natural history of AN disease and the frequent evolution from the AN-R to AN-BP subtype over time [26,27].

Few studies have evaluated the micronutrients (vitamins and trace-element) status in patients with AN. When comparing to literature findings, our patients were older and more severely undernourished. 

Zinc deficiency was the most frequent trace element deficiency that was observed in our study, this deficiency and its consequences are described in AN patients. Castro et al. [28] studied the biochemical and nutritional parameters at admission and at discharge in patients with an average BMI of 15; more than 80% of subjects had at admission low levels of zinc. In an experimental study, Suzuki and colleagues highlighted, in rates, an association between blood zinc levels and eating behavior regulation [16]. Indeed, the zinc role in the synaptic neurotransmission modulation of the GABAergique system, which is thought to act in the AN physiopathology, has been described [29]. In 1994, a randomized controlled trial of zinc supplementation in AN patients reported a two-fold increase of BMI in the zinc group [30]. A meta-analysis of zinc studies supported the evidence for the need of zinc supplementation in AN patients [31].

More than 20% of our patients had selenium deficiency. One retrospective study of 153 AN patients with an average BMI at 16.5 for AN-R and 18.8 for AN-BP, selenium deficiency was the most frequent, being observed in half of patients [32]. Moreover, AN-BP patients are exposed to cardiac complications because of associated ionic electrolyte disorders, such as hypokalemia [33] and hypomagnesaemia [34], potentially leading to arrhythmia and sudden death [35]. Our results seem to confirm this cardiac risk that may potentially be increased in AN-BP patients. Indeed, this group of patients had a significantly lower LVEF when compared to the AN-R patients. Selenium deficiencies may aggravate these cardiac complications [36]. Furthermore, Agnello and colleagues described selenium importance as an antioxidant system, cofactor of myocardial function, and regulator of both anxiety and mood. Chronic malnutrition is known to lead to antioxidant deficit [37]; these outcomes support a systematic selenium supplementation in AN-BP patients [38]. 

Thiamin deficiency is frequently described in alcohol use disorders and in patients after bariatric surgery with devastating neuro-cognitive consequences [39,40]. Frequent vomiting after bariatric surgery is described as a potential risk factor of thiamin deficiency [41]. In AN patients, Winston et al. described in 37 AN patients, among them, 38% had thiamin deficiency, but not related to food restriction duration, vomiting frequency, or alcohol consumption [42]. In our cohort, 15% of our patients had thiamin deficiency, but there were no neurological consequences, probably because a systematic supplementation is provided for all patients at admission. In our study 15% of patients had thiamin level upper the higher limit. It is explained by the fact that patients that were hospitalized in our unit suffer from chronic forms of AN, with several hospitalizations in intensive care unit and emergency rooms, with previous intravenous supplementations.

Thirty-seven percent of our patients had copper deficiency. Few studies have specifically focused on copper deficiency and its consequences in AN patients without relevant evidences [32]. Prospective studies are needed to explore the consequences of copper on arrhythmia, anaemia [43], neutropenia [44], and bone demineralization [45] in patients with AN.

Vitamin B9 and B12 deficiency was rare in our patients (9% and less than 5%, respectively). In the literature, vitamin B9 deficiency varies between 20% and 45% [28,32]. No large studies were found for vitamin B12 deficiency in AN patients. One published case report described sensory neuropathy consequences of vitamin B12 deficiency in one patient [17].

More than half of our patients had vitamin D deficiency. Davide Gatti found the same prevalence (58%) in a cohort of 89 AN patients, despite a higher average BMI at 15. A strong relationship between vitamin D deficiency and hip bone mineral density was found [46]. Association between hypovitaminosis D3 was found in a study by Tasegian et al. that might be associated to an inflammatory response deficit, and depressive symptoms in patients with long-term eating disorders. [47] Despite severe malnutrition, the bioavailability of oral ergocalciferol in young AN patients was similar to that of healthy controls [48]. Vitamin D deficiency can contribute to bone mineral density loss, which is the most frequent chronic complication of AN disease [49]. The supplementation of vitamin D should be provided to all AN malnourished patients [46]. Further study on nutrient intakes in women with AN-R when compared with a control group, suggested the benefit of vitamin D prescription for both groups to prevent osteoporosis [50].

Our highly specialized eating disorders unit receives patients with severely chronic malnutrition and in life threatening conditions. Often, they underwent micronutrients supplementation before hospital admission, including emergency or critical care units. This could explain the only 9% of folic acid deficiency and the high rates of vitamin B1 (28.9%) and B12 (30%) in our patients [32] when comparing to the literature. In accordance with our founding another study, with in 70 AN-R patients with an average BMI at 15, had an increase of both plasma vitamin B12 (38.5%) and vitamin B9 (4.3%) in AN patients [51].

In comparison to general population, in this study, malnourished anorexia nervosa patients were largely deficient in micronutrients. Indeed, in a large epidemiological study that was conducted by Hercberg et al. (1991), only 5% of the general population had a deficiency in Zinc and 14% of men with less than 50 years old presented a copper deficiency [52]. The SU.VI.M.AX study showed only 2% of low plasma concentration of selenium in adult volunteers [53]. Selenium deficiency is associated with a high risk of mortality and cognitive impairment, while zinc deficiency expose to a cognitive decline, neurological disorders, osteoporosis, and infections. These deficiencies in severely malnourished AN patients could provide additional comorbidities and aggravate their prognosis in medium and long term, hence a systematic unspecific micronutrient supplementation, as recommended by the French and American guidelines [54,55]. This approach does not address the targeted micronutrient supplementation that should be optimal. Still, more studies are needed to precise the optimal level of supplementation for each micronutrients, including cost/efficacy aspects. 

### Limitation

The determination of micronutrients and vitamins plasma concentrations in routine practice may not reflect their total body pool, leading to deficiencies underestimation, indeed micronutrients are mostly intracellular components. 

It is well known that some patients, including undernourished, were prescribed micronutrients supplements before admission, leading some measurements to be unreliable.

Finally, since all statistical analyses were conducted in exploratory manner, *p*-values that were smaller than 0.05 were considered as statistically significant without correction for the type I error inflation induced by multiple comparisons. Hence, the results should be interpreted with caution, as some p-values may only be significant by chance.

## 5. Conclusions

This large study cohort showed that severely malnourished AN patients have many micronutrient deficiencies; zinc and vitamin D are the most frequent, followed by copper, selenium, and vitamin B1. The blood levels of these deficiencies varied, depending on AN subtype. Thus, micronutrients status must be monitored and supplemented at the admission to avoid deficiencies that are related symptoms during renutrition. Prospective studies are needed to explore the symptoms and consequences of each deficiency, which can aggravate the prognosis during recovery.

## Figures and Tables

**Figure 1 nutrients-11-00792-f001:**
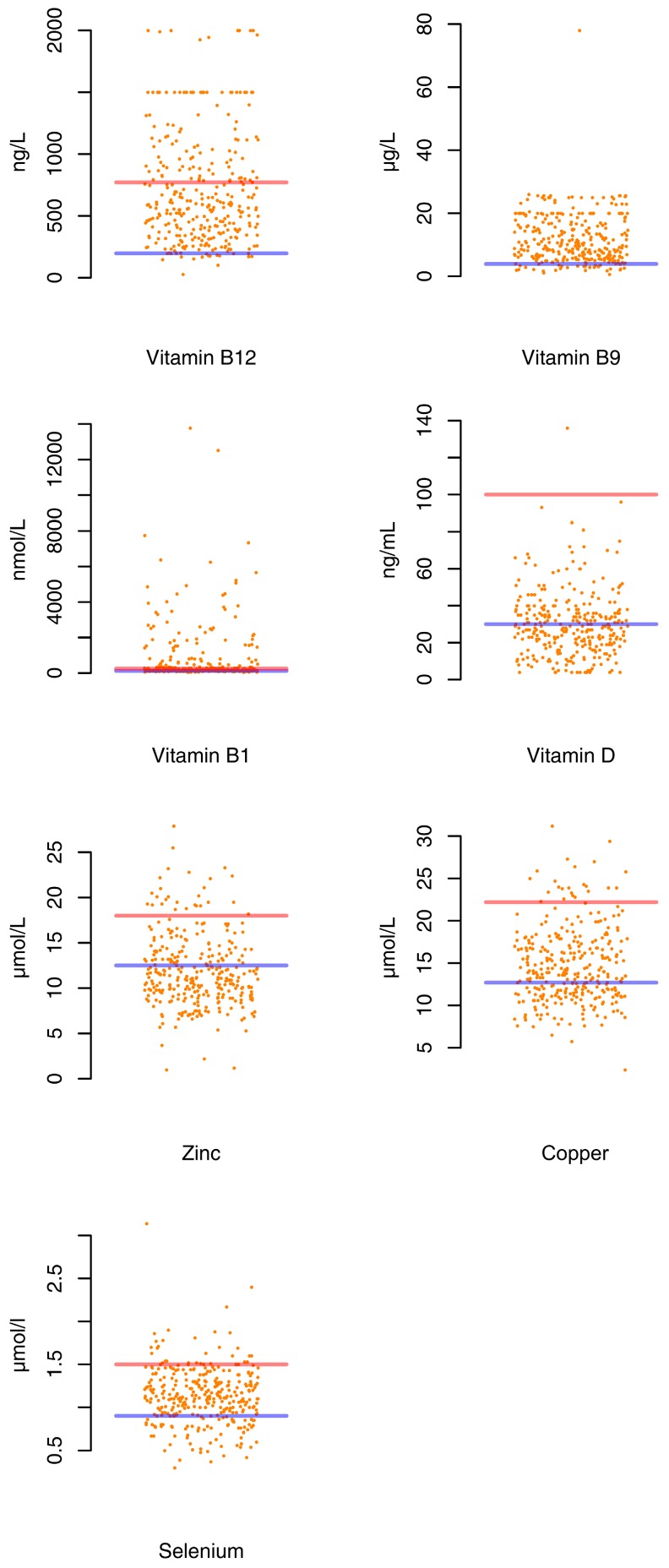
Micronutrients status in AN patients. Lower limits of normal micronutrients concentrations values are represented by blue lines and upper limits of normal values with redlines. Dots under the blue lines represent micronutrient deficiencies in AN patients.

**Table 1 nutrients-11-00792-t001:** Characteristics of all patients at admission and comparison between Anorexia Nervosa restricting type (AN-R) and AN binge-purging (AN-BP).

Clinical and Biological Parameters	*N*	AN-BP	AN-R	*P*
Patients (*n*, %)	374	121 (32%)	253 (68%)	
Gender	360 (96.3%) Women	117 (96.7%)	243 (96%)	NS
	14 (3.7%) Men	4 (3.3%)	10 (4.1%)	NS
Age (year)	31.3 ± 12.9	29,41 (23.73, 37.09)	26,22 (19.98, 38.29)	0.029
Weight (kg)	33.7 ± 5.9	34.40 (30.00, 39.10)	32,40 (29.60, 36.80)	0.048
Height (m)	1.6 ± 0.07	1.64 (1.60, 1.68)	1.62 (1.59, 1.68)	NS
BMI (18.5–24.9 kg/m2)	12.5 ± 1.7	12.80 (11.25, 14.20)	12.30 (11.30, 13.50)	NS
Fat Mass (g)	3098 ± 1744	2715.00 (2236.75, 4073.73)	2505.60 (2165.75, 3381.02)	NS
Fat Mass (%)	9702 ± 4057	8.60 (7.75, 12.45)	8.50 (7.40, 9.43)	NS
Disease duration (year)	9.4 ± 9.2	10.00 (5.00, 16.00)	4.50 (2.00, 12.00)	<0.001
Albumin (38–52 g/L)	37 ± 6.8	38.00 (32.00, 42.00)	37.75 (34.00, 41.00)	NS
CRP (<5 mg/L)	4 ± 14.5	0.71 (0.50, 1.20)	0.70 (0.50, 1.45)	NS
TSH (0.55–4.78 mUI/L)	1.9 ± 1.5	1.35 (0.90, 2.23)	1.62 (1.14, 2.35)	NS
Ca (2.12–2.52 mmol/L)	2.2 ± 0.17	2.25 (2.17, 2.38)	2.22 (2.14, 2.30)	NS
Ph (0.8–1.45 mmol/L)	1.2 ± 1.6	1.15 (0.94, 1.27)	1.15 (0.98, 1.31)	NS
AST (15–37 UI/L)	60.4 ± 116.3	28.00 (21.00, 38.00)	29.00 (21.00, 54.00)	NS
ALT (12–78 UI/L)	109 ± 232.8	36.00 (26.25, 65.75)	51.00 (30.00, 103.50)	0.001
GGT (5–55 UI/L)	63.1 ± 135.6	29.00 (18.00, 56.50)	34.00 (20.00, 59.00)	NS
ALP (46–116 UI/L)	90.2 ± 105.3	70.00 (52.75, 95.50)	70.00 (53.00, 94.00)	NS
BNP (<100 ng/L)	47.3 ± 68.4	20.00 (9.00, 46.00)	31.00 (15.00, 52.75)	0.021
LVEF (>50%)	64.87 ± 7708	65.00 (57.00, 67.00)	66.00 (61.00, 71.00)	0.009

Values are presented on median and Standard Deviation or on median and extremes; NS: not significant; BMI: body mass index; CRP: C reactive protein; TSH: thyroid balance; Ca: Calcium; Ph: Phosphoremia; AST: aspartate aminotransferase; ALT: alanine aminotransferase; GGT: gamma glutamyl transferase; ALP: alkaline phosphatase; BNP: brain natriuretic peptide; LVEF: left ventricular ejection fraction.

**Table 2 nutrients-11-00792-t002:** Deficiencies detected in AN patients.

Deficiencies Number	% of Patients
None deficiency	7.2%
One deficiency	28.3%
Two deficiency	33.2%
Three deficiency	18.8%
Four and more deficiency	12.6%

**Table 3 nutrients-11-00792-t003:** Comparison of micronutrients status between AN-R and AN-BP.

Micronutrients	AN-BP	AN-R	*p*
Zinc (12.5–18 micromol/L)	11.10 (9.05, 13.20)	11.45 (9.50, 14.07)	NS
Copper (12.7–22.2 micromol/L)	15.25 (11.75, 17.70)	13.50 (11.30, 16.72)	0.022
Selenium (0.9–1.5 micromol/L)	1.02 (0.83, 1.21)	1.18 (1.00, 1.38)	<0.001
Vitamin B12 (197–77 1ng/L)	516.00 (337.00, 804.00)	608.50 (394.25, 976.75)	0.036
Vitamin B9 (>3.90 microg/L)	9.32 (6.18, 14.70)	10.66 (6.19, 17.55)	NS
Vitamin D (30–100 ng/mL)	26.00 (15.75, 35.25)	29.00 (20.75, 36.25)	NS
Vitamin B1 (126–250 nmol/L)	188.50 (150.00, 334.25)	195.00 (148.00, 457.00)	NS

Values are presented on median (interquartile range). AN-R, anorexia nervosa-restricting subtype; AN-BP, anorexia nervosa-binge-purging subtype; NS, not significant.

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
