# Peer review of "Micronutrients Deficiencies in 374 Severely Malnourished Anorexia Nervosa Inpatients"

_nutrients, 2019, doi:10.3390/nu11040792_

Reviewer 1 Report

Summary

The authors investigate micronutrients deficiencies in patients with anorexia nervosa. Although the study design is straightforward, the relevance and benefit of the study is unclear. I therefore recommend a rejection of the manuscript.

 Major Issues

 1.   Relevance and benefit of the study are unclear. It is not new or unexpected that a patient group whose disorder is characterized by low body weight has micronutrients deficiencies. There are already some reviews covering this topic (e.g. Setnick, 2010), the authors themselves state that the supplementation of micronutrients is already recommended in treatment guidelines and are referencing about ten studies in the discussion section that investigated the same topic (with mostly the same results).

 2.   There are quite a few claims that are either not reported by references or plainly not correct, e.g.:

·           “undernutrition is frequent” in patients with AN (line 17): undernutrition is not only frequent but the very first diagnostic criterion, so the statement by the authors seems very misleading. The diagnostic criterion of underweight is also missing in the characterization of AN (lines 41-43).

·      “AN is a common and serious psychiatric disease” (line 41): This is not correct since AN is one of the rarest among the psychiatric disorders (the authors name the prevalence themselves a few lines later).

·      Lines 52-53: This is not supported by a reference.

·      The references in the discussion seem to be mixed up, e.g. line 184: the text refers to a recent study, the reference refers to the DSM. The references in the lines 190-204 also seem not to fit to what is reported in the text. The study by Tasegian et al (line 218) can not be found in the reference section at all.

 3.   The manuscript needs major editing of language and style as well as formatting issues, to give a few examples:

·                     Abbreviations in the abstract are not explained

·                     BMI is expressed as kg/m2 not kg/m2

·                     Line 26: “a lived longer”: unclear what this means

·                     Line 46-47: The sentence implies that 50% of patients with AN die of somatic

complications which was probably not what the authors wanted to say.

 Minor Issues

4.   The authors should address the found differences in patients with AN-R and AN-BN in light of the major difference of duration of the disorder in the discussion section.

5.   Results reported in the text, Table 2, Figures 2, 3 and 4 are redundant since they contain the same information presented differently visually. I would suggest cutting these down to reporting each result once.

6.   There are several paragraphs that contain only one sentence which is not in line with a scientific writing style.

 Author Response

Reviewer 1 Major Issues 1. Relevance and benefit of the study are unclear. It is not new or unexpected that a patient group whose disorder is characterized by low body weight has micronutrients deficiencies. There are already some reviews covering this topic (e.g. Setnick, 2010), the authors themselves state that the supplementation of micronutrients is already recommended in treatment guidelines and are referencing about ten studies in the discussion section that investigated the same topic (with mostly the same results). The reviewer is right, we made the following corrections: This study reports micronutrients deficiencies in a large cohort (in our knowledge the largest one) of severely malnourished adults patients (N: 374 average age: 27, BMI: 12.5). The others studies concerned smaller cohorts of less severely malnourished patients. Achamrah 2017; N: 153, average age: 28 and BMI: 17 Setnick, 2010 : review literature We aimed to investigate if in a more severe population, if deficiencies were more severe and more frequents. Our results confirm that. 2.   There are quite a few claims that are either not reported by references or plainly not correct, e.g.: -“undernutrition is frequent” in patients with AN (line 17): undernutrition is not only frequent but the very first diagnostic criterion, so the statement by the authors seems very misleading. Sorry for this mistake, the reviewer is right, we correct now: “Undernutrition, one of the majors diagnostic criteria of AN, can be associated with several micronutrients deficiencies “ (Lines 17- 18 page 1) -The diagnostic criterion of underweight is also missing in the characterization of AN (lines 41-43). “AN is characterized by a serious disturbed eating behaviours leading to underweight, together with distress or excessive concern about body shape or body weight and hyperactivity”. (Line 43 page 1) -“AN is a common and serious psychiatric disease” (line 41): This is not correct since AN is one of the rarest among the psychiatric disorders (the authors name the prevalence themselves a few lines later). Thank you for this comment, it is right we correct now: “Anorexia nervosa (AN) is a psychiatric disease characterized by a serious disturbed eating behaviors …” line 43 page 2 -Lines 52-53: This is not supported by a reference. We addressed your comments, we add citations: line 61 page 2 -The references in the discussion seem to be mixed up, e.g. line 184: the text refers to a recent study; the reference refers to the DSM. The references in the lines 190-204 also seem not to fit to what is reported in the text. The study by Tasegian et al (line 218) cannot be found in the reference section at all. Sorry for this, it was due to Zotero update; we correct now and check all bibliography order. 3.   The manuscript needs major editing of language and style as well as formatting issues, to give a few examples: Abbreviations in the abstract are not explained: Done BMI is expressed as kg/m2 not kg/m2: Done Line 26: “a lived longer”: unclear what this means, We changed the sentence Line 46-47: The sentence implies that 50% of patients with AN die of somatic complications which was probably not what the authors wanted to say. Sorry for this unclear sentence, we changed it know (lines 52-53page 2): “A mortality rate of 5-10% at ten years has been reported. Somatic complications and suicide are the two first mortality causes”. Minor Issues 4.   The authors should address the found differences in patients with AN-R and AN-BN in light of the major difference of duration of the disorder in the discussion section. The reviewer is right, we now add the following sentence: “This difference on disease duration could explain that the majority of micronutrients deficiencies, such as Zinc, Selenium and Vitamin B12 were more important in AN-BP patients.” (Page 9 line 215) 5.   Results reported in the text, Table 2, Figures 2, 3 and 4 are redundant since they contain the same information presented differently visually. I would suggest cutting these down to reporting each result once. We thank the reviewer for these comments, but each table or figure brings different information: The table 2 compare clinical and biological features between the two subtypes of AN, figure 2 exposes micronutrients deficiency prevalence in our cohort, while figure 3 illustrate the distribution of micronutrients status. Figure 4 exposes the numbers of micronutrients deficiencies by patients. If our answer is not satisficing, we propose to remove figure 4 6.   There are several paragraphs that contain only one sentence which is not in line with a scientific writing style. Yes that’s true, we correct.

Reviewer 2 Report

This paper presents data on micronutrient deficiencies in a large sample of severely underweight anorexia nervosa patients at admission to inpatient treatment. Zinc deficiency was most prevalent, followed by Vitamin D and Copper. Deficiencies in the remaining micronutrients were not particularly prevalent. The topic of this paper is interesting and relevant to the journal. However, I have several concerns and/or recommendations for this paper, as follows:

 Introduction: 

Most details are quite general to AN and details/justification for the research are quite vague. Please be more explicit - for example, it would be good to give examples of somatic complications described in the literature (line 50) and clinical and biological symptoms (line 52) and examples of how they can be explained by micronutrient deficiencies (line 53).

There is no explanation for why you would consider differences between AN subtypes and also why you would expect differences in micronutrient deficiencies between AN-R and AN-BP – this needs to be more fully described to add to the justification for your research design. 

I feel Table 1 is not needed and would recommend removing it. You have described the criteria/characteristics at the beginning of the introduction and so this information does not need to be repeated in table form. Also as you are including DSM IV and V, it is odd to provide specific details on DSM V only, given diagnostic criteria did change. However, I would recommend that you include a couple of sentences describing the differences between AN-R and AN-BP in the introduction - this would be important to do so, given that you are considering the subtypes separately.

Previous studies seem to have focused on AN patients with a higher BMI 15-16. I think it would be important to strongly emphasise this in the introduction (and also in the discussion) i.e. that you are assessing micronutrients in a severely more unwell patient group than previously studied and whether you would expect different findings.

Why did you choose to measure these specific micronutrients? 

 Method:

Figure 1 – I would recommend removing this figure as typically a flow chart is not needed unless it is an RCT. It also does not add much to the methods section and may be best being briefly described in a sentence in section 2.1.

Section 2.3. It may be more appropriate for this section to be called 'Measures and Procedure' as you include a lot of procedural information too.

Line 76 – should be ‘binge-purging type’

Line 85-93 biological data – I think it needs to be specified where the normal values are from/what population are they based on (especially as it is not a specially selected control group). Is this the reference measurements you refer to in line 80? Are the normal values based on an appropriate comparison group i.e. same gender/age etc.?  

Line 85-93 biological data – I would recommend removing the normal values from this paragraph and putting them in an additional column in the tables, so it is easy for the reader to see whether the medians you present fall in the normal range.

Line 85-93 biological data – There are several biological parameters presented here but it is not clear how all of them are relevant to the study question (e.g. CRP - an inflammatory marker, BNP - related to cardiovascular functioning) – are they all needed? How do they relate to the research question? What do they tell us about the sample? Perhaps consider whether all of them are needed and be more explicit as to what groups of biological parameters you are including and why you are reporting them. 

Line 94-95 – It is unclear how this variable (LVEF) is relevant to the study question as there is no mention of it in the introduction or discussion. Does this variable need to be included? What does it show and how is it relevant? 

Section 2.4 – It is unclear how this is relevant information for the current study, given that measurements were taken on admission before this treatment/refeeding began and this study does not report longitudinal findings. Should it be included?

Section 2.5 – Specify why you have used non-parametric tests (rather than parametric tests). Also as you statistically analyse many variables, I think it is important to correct for multiple testing, which you don’t seem to have done. The methods for doing this should be reported in section 2.5. If you choose not to correct for multiple testing, this needs to be explicitly stated in the ‘Limitations’ section.

 Results:

I would be interested to see the results for the AN group as a whole i.e. not broken down by subtype, given that the research you describe in the discussion does not seem to consider this and that a clear rationale has not been presented in the intro for considering these groups separately. Perhaps an additional column in the tables could be included.

Table 2 – I recommend reporting the fat mass variables (g and %) in the rows under BMI i.e. keep related variables together in the table e.g. anthropometric: weight, height, fat mass should be reported in subsequent rows.

Figure 2 – I would recommend removing this figure as you have nicely described the percentages in the first paragraph of section 3.2, so this information does not need to be repeated.

Figure 4 – I think the information in Figure 4 would be best described in a Table or in the text (rather than in a graph). It would be more informative to the reader.

Figure 3 – this is a great figure and very informative. I would recommend making the figures bigger, so the reader can better see the details, particularly as for Vitamin B1 and B9 we are unable to see the data points in the space between the upper and lower limits. Also there appears to be no red line on Vitamin B9 graph.

 Discussion:

Line 164-180 – the information regarding these extra nutritional parameters is interesting. However, I feel there is not enough information in the earlier sections of the paper (i.e. introduction, methods, results) to explain the relevance and importance of these in relation to micronutrient deficiencies i.e. how do these nutritional proteins relate to micronutrients and why are you reporting them, how do your findings with the proteins link with the micronutrient findings. I would recommend including more details throughout the paper on these additional nutritional parameters.

Line 174-176 – this doesn’t seem to fit into the discussion and seems a bit of a random comment – why is it important to the findings? Perhaps it could be expanded on or deleted.

Line 185-189 – I would recommend changing folic acid to Vitamin B9 in this section to keep it consistent with how you have named it.

Line 190 and 191 – references needed for both of these sentences.

Line 197-200 – this seems to be an important point but it does not flow on from the previous section. Perhaps comment on what you found in relation to selenium to introduce this paragraph?

Line 206 – I feel you have not provided enough background on the micronutrients for it to be clear to the reader (who may not know much about this topic e.g. if from an eating disorder background) why age and illness duration may make thiamine deficiency more frequent. I would recommend including some more information on the micronutrients (potentially in the introduction if it doesn't fit here).

Line 213-214 – The final point should be expanded on – e.g. how may this affect the study results? What does it mean for the current study? Why is this important for the current study?

Line 231-233 – reference needed for this sentence. Also, this information would make most sense to be separated and included in the paragraphs where you discuss zinc and selenium respectively.

In Figure 4 the B vitamins seem to have lots of people over the upper limit – perhaps this could be commented on – could this be due to previous supplements?

Most people only had one or two deficiencies i.e. the rest of the micronutrients were normal – it would be good to have a comment on this e.g. why this would be the case? Is this surprising given the levels of food restriction in these patients?

The median values do not seem to be that far below the normal range (e.g., for zinc) – how deficient are these patients? At what level would this start to cause problems?

Line 181-238 – I think the structure needs to be reworked in this section. Studies seem to be presented in a confusing order. One suggestion would be to discuss each micronutrient deficiency separately i.e. discuss your findings, previous findings in AN and general population (to put your findings into context of healthy people) and how the micronutrient may be involved in AN symptoms, behaviours and medical complaications separately for each micronutrient.

Limitations – it may be also important to include a limitation regarding the normal values. Are the normal values you present from women in the same range with a healthy BMI to ensure they are a good comparison? i.e. are the patients deficient in comparison to an appropriate comparison group? If not, then perhaps this should be commented on.  

Additional comments:

Abbreviations: Make sure to consistently use abbreviation for AN throughout e.g. line 226.

References in discussion: After the citation where the author has been named, please include the reference number so it is easy to find in the reference list.

Please have a native English speaker proofread this article – there are many issues with word usage, words missing and incorrect word endings.

Author Response

Reviewer 2 This paper presents data on micronutrient deficiencies in a large sample of severely underweight anorexia nervosa patients at admission to inpatient treatment. Zinc deficiency was most prevalent, followed by Vitamin D and Copper. Deficiencies in the remaining micronutrients were not particularly prevalent. The topic of this paper is interesting and relevant to the journal. However, I have several concerns and/or recommendations for this paper, as follows: Introduction: -Most details are quite general to AN and details/justification for the research are quite vague. Please be more explicit - for example, it would be good to give examples of somatic complications described in the literature (line 50) and clinical and biological symptoms (line 52) and examples of how they can be explained by micronutrient deficiencies (line 53). The reviewer is right, we correct now: Lines 58-61 page 2: “Cardiac failure risk, hypertransaminasemia and hepatic failure, functional intestinal disorders such us blotting and constipation, hematological disturbances, bone demineralization and hormonal disorders are the most frequents complications”. Lines 64 to 67 page 2: “For example, Suzuki et al described a link between zinc deficiency and restrictive eating behavior; a case of sensory neuropathy was found in AN patient with Vitamin B12 deficiency; fasting can cause neurological complications after severe vitamin B1 malnourishment.” -There is no explanation for why you would consider differences between AN subtypes and also why you would expect differences in micronutrient deficiencies between AN-R and AN-BP – this needs to be more fully described to add to the justification for your research design. We chose to compare between the two subtypes of AN because we expected the hypothesis that patients with AN Binge-Purging type, in addition to poor oral intakes, may have a different and more severe deficiencies due to vomiting and laxative misuses . We add the following sentence lines 67 to 69 : “Differences may be expected between the two subtypes of AN because of purging behaviors which can lead to more severe and different micronutrients deficiencies due to high and low digestive loss. -I feel Table 1 is not needed and would recommend removing it. You have described the criteria/characteristics at the beginning of the introduction and so this information does not need to be repeated in table form. Also as you are including DSM IV and V, it is odd to provide specific details on DSM V only, given diagnostic criteria did change. Ok we remove table 1 -However, I would recommend that you include a couple of sentences describing the differences between AN-R and AN-BP in the introduction - this would be important to do so, given that you are considering the subtypes separately. The reviewer is right we add this sentence lines 45 to 48 page 2 : “ AN is found in two major types: the restricting type (ANR) in which patients limit food consumption and, binge-eating/purging type (ANBP) , in which patients in addition to food limitation, exhibit cycles of large meals followed by purging behaviors (vomiting and/ or laxative abuse) “ -Previous studies seem to have focused on AN patients with a higher BMI 15-16. I think it would be important to strongly emphasize this in the introduction (and also in the discussion) i.e. that you are assessing micronutrients in a severely more unwell patient group than previously studied and whether you would expect different findings. Line 61 to 64 page 2: “However prevalence of micronutrient deficiency and its consequences in malnourished patients with AN are poorly known and available studies focused in patients with moderate undernutrition (Mean BMI > 15).  Our study focused for the first time in our knowledge on very severely malnourished adult patients with severe and chronic forms of AN -Why did you choose to measure these specific micronutrients? We chose especially these micronutrients because they were the most frequently described in the literature and in our study we described all micronutrients dosed during usual medical care. Method: Figure 1 – I would recommend removing this figure as typically a flow chart is not needed unless it is an RCT. It also does not add much to the methods section and may be best being briefly described in a sentence in section 2.1. We are ok with that, we remove it. We add this description lines 82 to 84page 2: “Three hundred ninety-four patients were hospitalized on the selected period, 17 refused to participate. Three patients were excluded for Celiac and Crohn disease so a total of 374 patients were included.” However if editorial board prefer maintaining the flow chart, we also agree with that. -Section 2.3. It may be more appropriate for this section to be called 'Measures and Procedure' as you include a lot of procedural information too. Done -Line 76 – should be ‘binge-purging type’ Done -Line 85-93 biological data – I think it needs to be specified where the normal values are from/what population are they based on (especially as it is not a specially selected control group). Is this the reference measurements you refer to in line 80? Are the normal values based on an appropriate comparison group i.e. same gender/age etc.?   The lines 80 and 82 said where the normal values are from. We add this sentence to the manuscript : Line 100-103 page 3 : “The normal values of the laboratory were previously determined by used of very large samples of healthy population including both sex and equivalent number of man and woman in adults human people”. -Line 85-93 biological data – I would recommend removing the normal values from this paragraph and putting them in an additional column in the tables, so it is easy for the reader to see whether the medians you present fall in the normal range. Done -Line 85-93 biological data – There are several biological parameters presented here but it is not clear how all of them are relevant to the study question (e.g. CRP - an inflammatory marker, BNP - related to cardiovascular functioning) – are they all needed? How do they relate to the research question? What do they tell us about the sample? Perhaps consider whether all of them are needed and be more explicit as to what groups of biological parameters you are including and why you are reporting them. We agree with this recommendation: for example some biological parameters are relevant to our study: CRP to detect inflammation, which can influence the level of selenium, vitamin B9 and vitamin B1 ; BNP which is a marker of cardiovascular failure (associated with selenium deficiency); GGT and ALP are markers of cholestasis (which can change level of blood copper); according with the proposition of the review we remove the TSC and citrulline as non-relevant biological marker. -Line 94-95 – It is unclear how this variable (LVEF) is relevant to the study question as there is no mention of it in the introduction or discussion. Does this variable need to be included? What does it show and how is it relevant? We included initially LVEF and BNP because we searched a correlation with micronutrients deficiency, especially selenium. None association was found finally. If the reviewer agrees, we can add this sentence in the results: lines 173-176 page 6 “None association was found between left ventricular cardiac function (LVEF, BNP) and micronutrients deficiency. However, even if the mean of LVEF was in the normal range for the two subtypes of AN, a significant lower LVEF in AN-BP sub-type was found (p=0.009).” And in the discussion: line 241 page 9: “Our results confirm this risk as well as LVEF was significantly lower in ANBP patients “ -Section 2.4 – It is unclear how this is relevant information for the current study, given that measurements were taken on admission before this treatment/refeeding began and this study does not report longitudinal findings. Should it be included? In fact the procedure of patient’s management is not use for the present study because the study is not a longitudinal follow-up. For this reason we suppress this paragraph -Section 2.5 – Specify why you have used non-parametric tests (rather than parametric tests). Also as you statistically analyses many variables, I think it is important to correct for multiple testing, which you don’t seem to have done. The methods for doing this should be reported in section 2.5. If you choose not to correct for multiple testing, this needs to be explicitly stated in the ‘Limitations’ section. “Statistical analyses were conducted in exploratory fashion without correction for multiple comparisons and a p value lower than 0.05 was thus consider to be statistically significant. Non parametric test (i.e.Wilcoxon rank-sum test and Fisher test) were used for comparing groups on continuous and categorical variables respectively in order to perform statistical inferences free of distributional assumptions and robust to outlying measurement in the data that made any gaussian assumption questionable.” Results: -I would be interested to see the results for the AN group as a whole i.e. not broken down by subtype, given that the research you describe in the discussion does not seem to consider this and that a clear rationale has not been presented in the intro for considering these groups separately. Perhaps an additional column in the tables could be included. Clinical and biological characteristics of patients Age (years) 31,3 + 12,9 Weight (Kg) 33,7 + 5,9 Height (cm) 1,6 + 0,07 BMI (kg/m2) 12,5 +1,7 Fat Mass (g) 3098 + 1744 Fat Mass (%) 9702 + 4057 Disease duration 9,4 +9,2 Albumin (g/l) 37 + 6,8 CRP (mg/l) 4 + 14,5 BNP (ng/l) 47,3 + 68,4 TSH (mUI/l) 1,9 + 1,5 Ph (mmol/l) 1,2 + 1,6 Ca (mmol/L) 2,2+0,17 AST (UI/l) 60,4 + 116,3 ALT (UI/l) 109 + 232,8 GGT (UI/l) 63,1 + 135,6 PAL (UI/l) 90,2 + 105,3 LVEF (%) 64,87 + 7708 Values are or median and SD; BMI, body mass index; CRP, the C reactive protein; TSH, thyroid balance; Ca, Calcium; Ph, Phosphoremia;  AST, aspartate aminotransferase; ALT, alanine aminotransferase; GGT, gamma glutamyl transferase; ALP, alkaline phosphatase ; BNP, brain natriuretic peptide; LVEF, left ventricular ejection fraction -Table 2 – I recommend reporting the fat mass variables (g and %) in the rows under BMI i.e. keep related variables together in the table e.g. anthropometric: weight, height, fat mass should be reported in subsequent rows. Done. -Figure 2 – I would recommend removing this figure as you have nicely described the percentages in the first paragraph of section 3.2, so this information does not need to be repeated. Ok the reviewer is right, we remove it -Figure 4 – I think the information in Figure 4 would be best described in a Table or in the text (rather than in a graph). It would be more informative to the reader. I agree with this recommendation; a short table is better than a graphic DEFICIENCIES NUMBER % OF PATIENTS None deficiency 7,2% One deficiency 28,3% Two deficiency 33,2% Three deficiency 18,8% Four and more deficiency 12,6% Ok we do that in a table Figure 3 – this is a great figure and very informative. I would recommend making the figures bigger, so the reader can better see the details, particularly as for Vitamin B1 and B9 we are unable to see the data points in the space between the upper and lower limits. Also there appears to be no red line on Vitamin B9 graph. Ok, we do that Discussion: -Line 164-180 – the information regarding these extra nutritional parameters is interesting. However, I feel there is not enough information in the earlier sections of the paper (i.e. introduction, methods, results) to explain the relevance and importance of these in relation to micronutrient deficiencies i.e. how do these nutritional proteins relate to micronutrients and why are you reporting them, how do your findings with the proteins link with the micronutrient findings. I would recommend including more details throughout the paper on these additional nutritional parameters. However nobody knows what is the status in micronutrients (blood level) in very severely adaptive state of malnourished, marasmus type AN patients with normal level of usual nutritional proteins marker. Despite un adaptive severely malnutrition and normal markers we have deficiencies in micronutrients. -Line 174-176 – this doesn’t seem to fit into the discussion and seems a bit of a random comment – why is it important to the findings? Perhaps it could be expanded on or deleted. We add know one sentence (Line 215, page 9) to make the link with micronutrient deficiencies: “This difference on disease duration could explain that the majority of micronutrients deficiencies, such as Zinc, Selenium ant Vit B12 were more important in AN-BP patients.” But if the reviewer prefer, we can remove it. -Line 185-189 – I would recommend changing folic acid to Vitamin B9 in this section to keep it consistent with how you have named it. Done -Line 190 and 191 – references needed for both of these sentences. Done -Line 197-200 – this seems to be an important point but it does not flow on from the previous section. Perhaps comment on what you found in relation to selenium to introduce this paragraph? The reviewer is right we edit the sentence lines 239page 9 “More than 20% of our patients had selenium deficiency; AN-BP patients are moreover exposed to cardiac complications because of associated ionic electrolyte disorders such as Hypokalemia and Hypomagnesaemia leading to arrhythmia and sudden death. Our results confirm this risk as well as LVEF was significantly lower in ANBP patients. The selenium deficiencies could aggravate these cardiac complications with myocardial failure” -Line 206 – I feel you have not provided enough background on the micronutrients for it to be clear to the reader (who may not know much about this topic e.g. if from an eating disorder background) why age and illness duration may make thiamine deficiency more frequent. I would recommend including some more information on the micronutrients (potentially in the introduction if it doesn't fit here). The reviewer is right we edit the sentence lines 251 to 257 page 10 “Thiamin deficiency is frequently described in alcohol use disorders and in patients after bariatric surgery with devastating neuro-cognitive consequences. Frequent vomiting could be a potential risk factor. In AN patients, Winston AP et al described in 37 AN patients, 38% of thiamin deficiency, but deficiency was not related to duration of eating restraint, frequency of vomiting, or alcohol consumption... In our cohort there was no neurological consequences, probably because, a systematic supplementation is provided for all patients at admission. Lines 262-264 page 10: Thirty-seven % of our patients had copper deficiency. Data are limited in the literature for copper status in patient with AN. Prospective studies are needed to explore the consequences of copper on arrhythmia, neutropenia and bone demineralization in patients with AN” -Line 213-214 – The final point should be expanded on – e.g. how may this affect the study results? What does it mean for the current study? Why is this important for the current study? Thanks you for this comment; in fact this final point was not relevant with the current study. We suppress the sentence. -Line 231-233 – reference needed for this sentence. Also, this information would make most sense to be separated and included in the paragraphs where you discuss zinc and selenium respectively. Done -In Figure 4 the B vitamins seem to have lots of people over the upper limit – perhaps this could be commented on – could this be due to previous supplements? Thank you for this comment, we add this sentence Line 261-264 page 9: “In our study % of patients had Thiamin level upper the Higher limit. It is explained by the fact that patients hospitalized in our unit suffer from chronic forms of AN, with several hospitalizations in intensive care unit and emergency rooms, with previous intravenous supplementations” -Most people only had one or two deficiencies i.e. the rest of the micronutrients were normal – it would be good to have a comment on this e.g. why this would be the case? Is this surprising given the levels of food restriction in these patients? It could be surprising that most patients present only one or two micronutrients deficiencies. We could expect higher percentage of patients with micronutrients deficiencies in this situation of extreme severely marasmus undernutrition, but in fact this situation is an adaptive form of malnutrition with no more proteins biological markers. Micronutrients needs in these situations are probably lower but it could increase at the recovery period. -The median values do not seem to be that far below the normal range (e.g., for zinc) – how deficient are these patients? At what level would this start to cause problems? Thanks for this comment For zinc for example the median in ANBP is at 11.10 micromol/L. This fact means that 50% of patients add level blood zinc under 11.10 and as the low level of normal value in the laboratory is 12.5 µmol/L it means that more than 50% of patients present zinc deficiency (low to 12.5 µmol/L). In fact 64.3% had a level of zinc in accordance with our result In fact low blood level in zinc or others micronutrients does not systematically means that this is a situation associated with micronutrients deficiencies but it is probably a higher risk to damask severely deficiencies during recovery. -Line 181-238 – I think the structure needs to be reworked in this section. Studies seem to be presented in a confusing order. One suggestion would be to discuss each micronutrient deficiency separately i.e. discuss your findings, previous findings in AN and general population (to put your findings into context of healthy people) and how the micronutrient may be involved in AN symptoms, behaviors and medical complications separately for each micronutrient. -Limitations – it may be also important to include a limitation regarding the normal values. Are the normal values you present from women in the same range with a healthy BMI to ensure they are a good comparison? i.e. are the patients deficient in comparison to an appropriate comparison group? If not, then perhaps this should be commented on.   The review is right. There is a limitation regarding how the normal values are determined. More often normal value are established by manufactory, some other scientific or physicist or laboratories prefer to establish normal values with a very large simple of normal health subject in normal population. In our university hospital it was the way to determine the normal value Additional comments: Abbreviations: Make sure to consistently use abbreviation for AN throughout e.g. line 226. Done References in discussion: After the citation where the author has been named, please include the reference number so it is easy to find in the reference list. Please have a native English speaker proofread this article – there are many issues with word usage, words missing and incorrect word endings.

Round  2

Reviewer 1 Report

The authors improved the introduction section, but a lot of my other comments were not addressed adequately. I can therefore not recommend publication at this stage. Some of the problems that remain:

- Rationale for the study was improved, but it is still unclear what the study adds to already existing literature. The key messages remain unclear (recommendations of treatment guidelines that are already in place are merely repeated in the conlusions section). 

- English language and style remain a big problem, there are still a lot of sentences that need to be greatly imrpoved. Otherwise, the reader will not be able to understand some sentences correctly.

- The citation the authors claimed to have added (page 2 line 61) is not there.

- Citations are still an issue, e.g. in the discussion section, p. 7, l. 179: this citation referes to the DSM-IV-TR manual? Furthermore the sentences in this paragraph are not a logical sequence and seem to randomly be together in one paragraph.

- Changes do not match with the pages and lines given in the author's reply making it very time-consuming for me to determine where the changes are.

- The discussion section needs more clarity, logical structure and focus. It is unclear what the reader should learn from this.

Author Response

-The authors improved the introduction section, but a lot of my other comments were not addressed adequately. I can therefore not recommend publication at this stage. Some of the problems that remain: 

Rationale for the study was improved, but it is still unclear what the study adds to already existing literature. The key messages remain unclear (recommendations of treatment guidelines that are already in place are merely repeated in the conclusions section). 

The reviewver is right, we clarified the introduction and improved the rational at the end of the introduction : Line 71-82 Page 2 

-English language and style remain a big problem; there are still a lot of sentences that need to be greatly improved. Otherwise, the reader will not be able to understand some sentences correctly.

We corrected and clarified english style 

-The citation the authors claimed to have added (page 2 line 61) is not there.

We add now line 61 page 2, citations 14 and 15 

-Citations are still an issue, e.g. in the discussion section, p. 7, l. 179: this citation refers to the DSM-IV-TR manual? Furthermore the sentences in this paragraph are not a logical sequence and seem to randomly be together in one paragraph. 

Done, we correct all the bibliografy

-Changes do not match with the pages and lines given in the author's reply making it very time-consuming for me to determine where the changes are.

We are sorry for that, we correct now

-The discussion section needs more clarity, logical structure and focus. It is unclear what the reader should learn from this. 

Done. We modified and corrected all the discussion part

Reviewer 2 Report

The authors have made substantial changes to the manuscript and it is much improved. I have some remaining concerns.

My main issue is that the English language is still incorrect in many places, particularly in the additions made to the manuscript. Please have a native English speaker proofread this article.

-The first paragraph of the introduction is too long - divide appropriately into 2. 

-Throughout, particularly in introduction and discussion: the additions you have made are useful but I feel some may not have been added in the best places - perhaps reread to check the order of information flows in a logical order. 

-The introduction does still not provide any background about the additional related parameters and nutritional proteins you included e.g. LVEF. Why have you included them as they don't link to your aim? Justify why you are looking at them. This was mentioned in your reviewers response but has not been included in the manuscript.

- The author explained to me in the report why certain micronutrients were selected but this should be explained in the manuscript too.

-Line 59: "blotting" should this be "bloating"

-Line 96-97: I think this sentence may go best after describing the aim in the next paragraph. Also what is "digestive loss" - this doesn't make sense.

-Section 2.1 page 2. You have not spelled out the abbreviation of DSM and have not included references to these.

-The new Table on page 4/5 is not needed - In my previous review I suggested it was incorporated into the existing table (now Table 2). I.e. add a column in for whole sample.

- Table 2: You have not included the units of measurement for the normal values. As you have reported the normal values here, I do not think it is necessary to include them in the section of 109 as well - rather direct the reader to the Table.

-Line 219 page 9: The addition of this sentence does still not explain why illness duration may be have an effect. Please explain.

-Line 238- 242: This section does not make sense - I think it needs to be reordered and slightly reworded.

-Line 254-255: Can you make this conclusion? Did you test for neurological consequences? If not, this needs to be removed.

- As said previously, as you have not controlled for multiple testing, this needs to be repeated in the limitations section. This is an important limitation.

- Please address this comment: References in discussion: After the citation where the author has been named e.g. Castro et al., please include the reference number so it is easy to find in the reference list i.e. Castro et al. [16].

Author Response

The authors have made substantial changes to the manuscript and it is much improved. I have some remaining concerns.

-My main issue is that the English language is still incorrect in many places, particularly in the additions made to the manuscript. Please have a native English speaker proofread this article.

We are sorry for that, We correct and clarified english style

-The first paragraph of the introduction is too long - divide appropriately into 2. 

Done

-Throughout, particularly in introduction and discussion: the additions you have made are useful but I feel some may not have been added in the best places - perhaps reread to check the order of information flows in a logical order. 

The reviwver is right we rereed and modified the introduction as well as the discussion.

-The introduction does still not provide any background about the additional related parameters and nutritional proteins you included e.g. LVEF. Why have you included them as they don't link to your aim? Justify why you are looking at them. This was mentioned in your reviewers response but has not been included in the manuscript.

We added it at the and of the introduction page 3 lines 85-89

-The author explained to me in the report why certain micronutrients were selected but this should be explained in the manuscript too.

Done, lines 83-85 page 3

-Line 59: "blotting" should this be "bloating"

Sorry we correct it

-Line 96-97: I think this sentence may go best after describing the aim in the next paragraph. Also what is "digestive loss" - this doesn't make sense.

Done, we correct now. Line 68-70 page 2

-Section 2.1 page 2. You have not spelled out the abbreviation of DSM and have not included references to these.

Sorry, we add citation of DSM 5 : Line 94, page 3, citation 4

-The new Table on page 4/5 is not needed - In my previous review I suggested it was incorporated into the existing table (now Table 2). I.e. add a column in for whole sample.

Done on table 1

-Table 2: You have not included the units of measurement for the normal values. As you have reported the normal values here, I do not think it is necessary to include them in the section of 109 as well - rather direct the reader to the Table. 

Done, we remove the biological data on the paragraph and we add the normal values on table 1 and table 2

-Line 219 page 9: The addition of this sentence does still not explain why illness duration may be have an effect. Please explain. 200-202

We remove the sentence

-Line 238- 242: This section does not make sense - I think it needs to be reordered and slightly reworded.

Done

-Line 254-255: Can you make this conclusion? Did you test for neurological consequences? If not, this needs to be removed.

We modified the sentence: page 9 lines 223-225

As said previously, as you have not controlled for multiple testing, this needs to be repeated in the limitations section. This is an important limitation.

Done we add this sentence on the limitations part: “Finally, since all statistical analyses were conducted in exploratory manner, p-values smaller than 0.05 were considered as statistically significant without correction for the type I error inflation induced by multiple comparisons.Hence, results should be interpreted with caution as some p-values may be significant only by chance”.

- Please address this comment: References in discussion: After the citation where the author has been named e.g. Castro et al., please include the reference number so it is easy to find in the reference list i.e. Castro et al. [16]

We checked all the bibliography
